# The Psychological Process of Residents’ Acceptance of Local Shale Gas Exploitation in China

**DOI:** 10.3390/ijerph17186736

**Published:** 2020-09-16

**Authors:** Liuyang Yao, Dangchen Sui, Xiaotong Liu, Hui Fan

**Affiliations:** 1International Business School/China, Shaanxi Normal University, Xi’an 710119, China; yaoliuyang@snnu.edu.cn (L.Y.); suihu7672@snnu.edu.cn (D.S.); fanhui@snnu.edu.cn (H.F.); 2College of Business/China Research Center for Social Entrepreneurship, Shanghai University of Finance and Economics, Shanghai 200433, China

**Keywords:** shale gas, acceptance, risk perception, benefit perception, structural equation modeling

## Abstract

Local communities and their opinion on shale gas exploitation (SGE) play an essential role in the implementation of energy policies, while little is known about the reasoning process underpinning the acceptance of SGE. The present study develops a conceptual framework to examine the psychological process of residents’ acceptance of local SGE, in which the impacts of trust, knowledge, and fairness are mediated by risk and benefit perceptions. Structural equation modeling has been applied to analyze the hypothesized relationships based on a dataset of 825 households in China’s largest shale gas field. Our results indicate that residents’ perceived fairness and trust positively affect their benefit perceptions and negatively affect their risk perceptions, which results in positive influences on acceptance, and knowledge of SGE’s environmental impacts positively affects perceived risks, which results in a negative influence on acceptance. Moreover, residents’ acceptance is primarily determined by their benefit perception, followed by perceived fairness, and knowledge is the least important determinant. Thus, our study contributes to the literature by exploring the structural relationships between various psychological predictors and the acceptance toward SGE, and the results from our empirical survey provide insight into designing appropriate strategies in the process of generating and communicating shale policies.

## 1. Introduction

Technological progress in horizontal drilling and hydraulic fracturing has dramatically expanded the commercial exploitation of shale gas and raised doubts about the sustainability implications on environmental, social, and economic development. On the positive side, shale gas exploitation (SGE) offers the promise for promoting economic growth, enhancing the security of domestic energy supply, and even facilitating the transition to a clean energy future [1]. On the negative side, SGE through hydraulic fracturing has generated enormous controversy considering the potential negative externalities on public health and the environment close to the exploitation areas [2]. Considering these trade-offs in SGE, shale policies across different countries and regions have moved in very different directions as to whether or not to embrace this new resource [3,4]. Thus, a careful understanding of residents’ acceptance of SGE in this case, as we will explore, not only helps to generate proper governance of the potential risks posed by SGE but also provides insight into making publicly acceptable shale policies.

Public perceptions of SGE have been analyzed in the literature for over 20 years, most of which are conducted in North America and European countries (for reviews, see Goldstein et al. [5], Thomas et al. [6], and Whitmarsh et al. [7]). However, the opinion on SGE may vary across geographic regions because of the vastly different backgrounds in culture and regulation [4]. Notably, this is the case of China, where investments into SGE and production are encouraged and subsidized by the central government. Unlike the fracturing debate in the Western countries, media coverage in China has primarily focused on SGE’s technological progress and benefits instead of its environmental risks [8,9]. Moreover, as most of China’s shale gas fields are located close to densely populated areas [10], where residents’ livelihoods are often tightly bound to the availability and utilization of natural resources, SGE in China may pose more pronounced environmental and health risks than it may do in the Western countries. At the same time, the state-owned Chinese mining companies may get preferential treatment at the local level, which gives the companies a temptation to downplay long-term environmental considerations [4,11,12]. Hence, public perceptions of SGE in China, particularly in the affected areas, could differ significantly from those investigated in other parts of the world.

China sits atop the world’s largest technologically recoverable shale gas reserves, while it did not start issuing its mining licenses until 2011. Along with several incentive policies in the ensuing years, the country has seen a steady ramp-up in yearly shale gas output, has become the world’s third-largest shale gas producer after the US and Canada, and currently seeks to achieve the ambitious production target of 30 billion cubic meters (bcm) by 2020 and 80–100 bcm by 2030. However, numerous challenges, including local opposition and the “not in my backyard” syndrome, have slowed the progress of China’s shale projects [9,13]. Thus, residents’ acceptance of SGE has been a critical factor for China to achieve the ambitious production goals. To date, however, only a few studies have been conducted to investigate public perceptions towards SGE in China [9,10,14,15,16,17] compared with that in the Western countries, and no studies have explored the psychological process and mechanisms that give rise to the acceptance of SGE.

This study aims to fill the research gap by examining the psychological determinants that lead to residents’ acceptance of local SGE in China. Specifically, residents’ acceptance of SGE has been explained as the result of a reasoning mechanism by considering the mediation roles of risk and benefit perceptions on the impacts of trust, knowledge, and fairness. Subsequently, given the latent structure of our constructed psychological constructs, the proposed conceptual framework has been tested using a structural equation modeling based on a cross-sectional dataset of 825 respondents in Southwest China. Lastly, with insights into residents’ psychological process of acceptance, we also discuss the development of appropriate strategies in the process of generating and communicating shale policies.

## 2. Theoretical Framework

Within the literature, there is a greater emphasis on the use of perception data to explain the motivations underlying the public acceptance of SGE [3]. Most of the empirical analysis investigated the acceptance of nationwide SGE that was not necessarily affecting their residential areas, such as Whitmarsh et al. [7] and Clarke et al. [18]. Our study contributes to the literature by focusing on the acceptance of people living in ongoing SGE areas who are constantly exposed to issues related to shale gas activities from their backyard. Moreover, Seigo et al. [19] made a distinction between expressed acceptance and revealed acceptance in the research of sustainable energy technologies. Expressed acceptance is related to the favorable attitude towards applying energy technologies, while revealed acceptance is the positive behavior in adopting energy technologies. As Sher [20] found that residents in China often had scruples in taking direct actions on local SGE due to the fear of getting themselves into trouble, residents’ expressed acceptance will be more suitable to reflect their honest feelings about local SGE in our study conducted in Southwest China.

Advances in the social sciences have identified a complex suite of psychological factors determining the acceptance of SGE, and we have developed a conceptual model (see Figure 1) to characterize the reasoning process underlying residents’ acceptance. This theoretical framework is inspired by the energy technology acceptance framework [19,21,22] and its subsequent empirical studies [23,24,25]. Our framework indicates that an individual’s decision about acceptance is primarily determined by risk and benefit perceptions, which are, in turn, influenced by several attitudinal factors, including knowledge, trust, and fairness.

### 2.1. Perceived Fairness

There are different types of fairness, and most empirical studies have focused on the influences of procedural fairness (i.e., procedures for decision-making) and distributional fairness (i.e., the distribution of benefits and burdens) on public acceptance [23]. With regard to SGE, Clough [26] and Besley [27] proposed that whether an individual’s view would be taken seriously by decision-makers (defined as interpersonal fairness) and whether an individual would be recognized as legitimate (defined as recognition fairness) should also be considered as important dimensions of just SGE. Considering that different types of fairness are closely connected in perception [28,29], this study uses the “perceived fairness” as a latent factor to consist of all sub-dimensional fairness and measures each type of fairness separately.

The role of fairness has been extensively studied within energy technology acceptance studies, such as carbon sequestration [19], wind power [30], and nuclear power [31]. However, no empirical studies with the qualitative method have analyzed the influence of fairness concerning the SGE context. As the energy technology acceptance literature suggested, perceived fairness would influence the acceptance indirectly, via its connections with risk and benefit perceptions associated with the energy technology [8,32]. Thus, we hypothesize that residents will stand a better chance of perceiving fewer risks and more benefits if they perceive the SGE as fair, and the following hypotheses are proposed accordingly:

**Hypothesis** **1** **(H1).**
*Residents who perceived more fairness in energy policy decisions will perceive fewer risks associated with local SGE.*


**Hypothesis** **2** **(H2).**
*Residents who perceived more fairness in energy policy decisions will perceive more benefits associated with local SGE.*


### 2.2. Knowledge

Knowledge of the energy technology’s impacts, in general, has been considered as an important predictor of public support, and it has been constantly measured with the respondents’ self-reported knowledge rather than their subjective knowledge [19]. The shale-related literature has examined the causality between knowledge and acceptance but has come to vastly different results. A simple correlational analysis by Stedman et al. [33] found that public knowledge of SGE was associated with more support in the UK sample, while it had no influence on public support in the US sample. Whitmarsh et al. [7] conducted an online survey of the UK public and also found more knowledge was associated with more favorable attitudes of SGE using the linear regression analysis. Using samples from nationwide studies conducted in the US [34] and Spain [35], results from the regression analysis even showed that knowledge negatively influences the acceptance of SGE.

A point of departure for understanding these mixed findings is that the literature has modeled the relationship between knowledge and acceptance in an overly simplistic approach, in which additional factors are often difficult to be incorporated for more complex analyses (such as those involving mediation). According to the technology acceptance framework [21], residents’ knowledge may play some role in influencing their risk and benefit perceptions and indirectly impacts their acceptance of the technology. Moreover, whether increasing the knowledge will result in more acceptance or not is uncertain because of the existence of pervasive confirmation bias. In other words, residents’ prior attitudes toward SGE will be reinforced rather than changed by their seeking out only confirmatory evidence [7,36]. As a result, the residents’ perceived risks and benefits are likely to be polarized as their knowledge increases from processing the mixed information in a biased way. Based on such insights, we proposed that those with more knowledge of SGE’s environmental impacts will have higher—both risks and benefits—perceptions of SGE. Therefore, the following hypotheses are proposed:

**Hypothesis** **3** **(H3).**
*Residents who have more knowledge of SGE’s environmental impacts will perceive more risks associated with SGE.*


**Hypothesis** **4** **(H4).**
*Residents who have more knowledge of SGE’s environmental impacts will perceive more benefits associated with SGE.*


### 2.3. Trust

A popular definition of trust was proposed by Rousseau et al. [37]: “Trust is a psychological state with the extent to accept vulnerability based upon positive anticipation of the intentions or behavior of another.” With limited capacities to evaluate or manage the consequences of SGE, residents have to employ trust as a filter that takes a broad range of available information sources as the input and forms the basis for their affective responses towards SGE [38,39]. Thus, residents get their cues about the risk and benefit information from relevant actors who implement or regulate the SGE projects. Trust in these relevant actors should have a substantial effect on their risk and benefit perceptions, as well as their acceptance of SGE.

The theoretical framework provided by Huijts et al. [21] and Kânoğlu and Soytaş [40] indicated that residents with higher trust in relevant actors would have less perceived risks and more perceived benefits, which, in turn, they would be more likely to accept. Most empirical studies also modeled the relationship between trust and acceptance indirectly, through risk and benefit perceptions [19,21], and only a few studies explored the direct influence of trust on acceptance [40,41]. Empirical findings also indicate that trust in different groups or institutions would have different influences on risk and benefit perceptions, as well as acceptance. For example, Brasier et al. [42] found that residents’ perceived risks of SGE were negatively related to their trust in the natural gas industry but were unrelated to their trust in state regulatory agencies. Yu et al. [14] also found residents’ trust in the local authority had no significant influence on their perceived risks, while more trust in the petroleum company and the central government would result in fewer perceived risks.

In the Chinese context, empirical studies found that residents rely mainly on the information given by experts and regulators who are highly in favor of developing shale resources as a heuristic or alternative foundation for their opinions [13,14]. Various studies have highlighted the differences in trust levels between central and local governments [43], but we found a similarity between residents’ trust in the central government and the scientists related to SGE during the pre-survey. Thus, we did not include residents’ trust in the local government and combined residents’ trust in scientists and the central government in one question. Following the most frequently modeled paths, the relationship between trust and acceptance in this study is fully mediated through perceived risks and perceived benefits, and the following hypotheses are proposed accordingly:

**Hypothesis** **5** **(H5).**
*Residents who have more trust in scientists and higher authorities involved in SGE will perceive fewer risks associated with SGE.*


**Hypothesis** **6** **(H6).**
*Residents who have more trust in scientists and the central government involved in SGE will perceive more benefits associated with SGE.*


### 2.4. Perceived Benefits and Perceived Risks

Residents’ tendency to accept an energy technology is found to be directly influenced by a weighing up of its risk and benefit perceptions [19,21,40,44]. As a significant energy alternative, SGE has immediate benefits for domestic energy security and provides a clean-burning, efficient, and affordable energy resource that is good for the climate by reducing air pollutants and greenhouse gas emissions. In addition to the general benefits, SGE also helps to power local economies by adding employment opportunities, increasing tax revenues, and improving public infrastructures (see Sovacool [45] for a review). Along with the benefits, SGE is often simultaneously perceived as a threat to the affected residents on several fronts. The most commonly cited risks posed by SGE have been environmental or health problems, prominently including noise pollution, geologic hazards, ecosystem degradation, air emissions, water use, and water contamination (see Costa et al. [46] for a review).

Previous survey-based studies have investigated people’s risk and benefit perceptions of SGE. A strand of the literature has focused on comparing risk and benefit perceptions among the general public [7,18] or residents living near SGE facilities [14,42]. Another strand of the literature has examined the influences of risk and benefit perceptions on the acceptance of SGE and found a positive (negative) relationship between perceived benefits (risks) and acceptance [7,18]. However, the mediation roles of risk and benefit perceptions, which have received considerable attention in the technology acceptance studies, have not been examined in the shale-related literature. The following hypotheses are proposed accordingly:

**Hypothesis** **7** **(H7).**
*Residents’ perceptions of SGE’s risks to the local communities will negatively affect their acceptance of local SGE.*


**Hypothesis** **8** **(H8).**
*Residents’ perceptions of SGE’s benefits to local communities will positively affect their acceptance of local SGE.*


## 3. Survey Design and Data

### 3.1. Study Area

Our data was collected in China’s first and largest shale gas field, Fuling shale gas field (FSGF), located in the Chongqing Municipality (Figure 2). The field lies underneath most of Fuling District, as well as a good portion of Nanchuan District and Wulong District. Sinopec—the nation’s second-largest oil and gas producer—is currently operating FSGF as its flagship project in Southwest China. Encouraged by the central government’s subsidies and tax incentives, Sinopec plans to accomplish the goal of expanding FSGF’s annual production capacity to 10 bcm by 2020 in two phases. The first-phase construction of 5-bcm production capacity was from 2013 to 2015, which encompassed 268.2 km^2^ area with 194 bcm of proven reserves. The second-phase construction of 5-bcm production capacity, which encompassed a 291.4 km^2^ area with 239 bcm of proven reserves, started in 2016 and is planned to end in 2020. At the end of 2018, Sinopec had drilled 402 wells in FSGF and produced about 60% of China’s total shale gas output. In the future, Sinopec’s Fuling project will continue to play a leading role in China’s process of achieving its ambitious goal of producing 80 to 100 bcm of shale gas in 2030. Thus, our study will not only extend our understanding of the social acceptability in FSGF but also provide insights into residents’ emotional responses in other SGE areas of China.

China has vastly different cultural and political values than the Western world has, which calls for a better understanding of the public’s attitude towards SGE. On the one hand, the vast and rapid expansion of SGE will undoubtedly have an impact on residents living close to the development areas. Moreover, the impact has been aggravated by the geologic features in Southwest China, which are characterized by a typical karst topography with hills, mountains, and subterranean rivers. The fragile karst topography helps provide pathways for the transport of pollutants from accidental well leakages, and these pathways also have the potential to allow hydraulic fracturing fluids transfer from deep underlying formations to shallow water aquifers. Additionally, China’s revenue- and job-hungry local authorities attach great importance to fulfilling the target of expanding the production capacity. Thus, a combination of “central protectionism” of the state-owned company and the insufficient regulatory capacity in the local environmental protection department has increased the likelihood of shale companies’ noncompliance with the rigorous environmental regulations [47,48].

On the other hand, the mass media displays different levels of both alarmism and optimism in SGE’s coverage between China and other shale gas-producing countries. In the UK and the US, the mass media included many posts on both the pros and cons of SGE [4]. However, Chinese outlets mainly focus on reporting SGE’s various benefits, such as satisfying the domestic clean energy demand and promoting economic development [13]. Previous studies found that the positive and vigorous SGE propaganda in China helped develop a positive view of SGE among the public and foster community pride [10,15]. In this context, residents living in FSGF are more likely to express a lower perceived risk and to speak positively about SGE.

### 3.2. Measures and Materials

Table 1 details how each construct was operationalized in the questionnaire. The indicators adopted herein were validated in existing scales and were further tested and modified by our pilot interviews with residents in the study area. All questions about fairness, knowledge, trust, perceived benefits, and perceived risks were measured on a 5-point Likert-type scale, where the responses were codified ordinally with “strongly disagree” as point 1, “disagree” as point 2, “neither agree nor disagree” as point 3, “agree” as point 4, and “strongly agree” as point 5. Responses for the questions about acceptance were measured by “strongly oppose” as point 1, “oppose” as point 2, “neutral” as point 3, “support” as point 4, and “strongly support” as point 5.

After evaluating the respondents’ attitudes towards SGE, the survey questionnaire also contained questions regarding respondents’ demographic information, as Table 1 presented. Most importantly, our investigators used their mobile phones to show the respondents an animated video during the face-to-face interview. The animated video, lasting one-and-a-half minutes, gave a brief introduction of the SGE’s hydraulic fracturing process before the measurement questions. This was found to be an effective way that helped improve respondents’ interest in participating in our survey and prompted respondents to think carefully about their responses to our questions.

### 3.3. Sample and Data Collection

The sample used in this study was obtained from face-to-face interviews by eight well-trained graduate students from July to August in 2018. To ensure the representativeness of residents living close to the SGE areas, we selected four towns in the Chongqing Municipality with extensive development of shale resources, including Baitao Town and Jiaoshi Town located in the first-phase construction area, as well as Pingqiao Town and Jiangdong Town located in the second-phase construction area. The population size of our selected towns varied from 15,000 to 65,000 residents. Proportional to the population size, 18 communities from the targeted towns—2 to 6 communities in each town—were randomly selected. In each of the selected communities, a simple random sample of 70 households was selected based on the household roster in each community. Accordingly, a total of 1260 households were involved in our household survey. Of those, 371 households failed to finish our questionnaire, because they were not at home or too busy to cooperate during our investigation, and 64 returned questionnaires were dropped due to a lack of confidence in data quality. As a result, 825 valid observations—65.48% of the designed sample size—were available for further analysis.

The majority of respondents (88.85%) were male, because our information was collected at the household level, and investigators were instructed to interview the household head as much as possible. The average age of participants was 47.27, 15.39% of the participants had an education level beyond the level of compulsory nine-year schooling, and about three-quarters of the sampled households consisted of three to five persons. The demographic characteristics of our sample provided a snapshot of aging, lower levels of education, and family size preference in FSGF, which was consistent with the population characteristics presented in the study of Yu et al. [15] conducted in another SGE area of Southwest China. The ratio of our sample sizes in two construction areas (57.09/42.91) was approximately proportional to that of the total household sizes (54.09/45.91), which ensured the same representation in each construction area. The average self-reported yearly per-person income was 3.69 measured by the 7-point scale, which is lower than the official statistics of 13,781 yuan in 2017. This indicated that our respondents’ incomes were probably underreported, a common situation in field surveys [54]. Overall, the characteristics of our sample were in-line with that of the population in FSGF.

## 4. Data Analysis and Results

Structural equation modeling (SEM) (see Anderson and Gerbing [55] for a review) has been used to test the research hypotheses, as well as the reliability and validity of the measurements in this study. Anderson and Gerbing [55] and Bentler and Dudgeon [56] proposed SEM, which combined a measurement model with a path model, as a solution to account for minimizing measurement errors and simultaneously estimating the structural regression. Since their initial work, SEM has been widely used in the research of energy research and social science, particularly to study the public preference for energy facilities (e.g., [23,51,53]. According to Anderson and Gerbing [55], the confirmatory factor analysis (CFA) will be used to assess the reliability and validity of each construct before using SEM to test whether the data support our research hypotheses.

### 4.1. Measurement Model Test (Confirmatory Factor Analysis)

The first step of our data analysis is to perform the confirmatory factor analysis (CFA) in evaluating the reliability and validity of a set of measures (Table 2). All constructs in our study have been measured by three to four formative indicators, as proposed by Kline [57], and Table 3 presents the descriptive statistics of these indicators. Based on the covariance matrix among the indicators (see the covariance matrix and correlation matrix, Table A1 and Table A2 in our Appendix A Materials), the relevant statistical information obtained from each construct’s CFA is shown in Table 2.

Table 2 shows a low level of residents’ perceived fairness of SGE in FSGF. Especially for distributive and procedural fairness, the mean values of *x11* (2.30) and *x12* (1.75) indicate that the respondents were less satisfied with the distributive aspects and rarely pertained to the decision-making process. The mean values of *x13* (2.82) and *x14* (3.10), by contrast, reveal residents’ neutral attitudes towards recognition fairness and interpersonal fairness. In other words, residents considered that they had the opportunity to raise their concerns that had not been adequately addressed by the government, which was consistent with the study of Tan et al. [9]. The reason might be the political and management structure in local energy projects, where community leaders have the responsibility to hear residents’ needs, which provides an opportunity for residents to express their concerns. However, local cultural and political norms inhibit community leaders’ communication with the government agencies of higher authority, especially when the residents’ concerns could create an unfavorable impression of the community. As for the other two exogenous constructs (*Knowledge* and *Trust*), residents generally held a neutral attitude about their knowledge of SGE’s environmental impacts and a more positive attitude about their trust in scientists and the central government, except the suspiciousness in environmental information disclosure regarding SGE. This might be because most of the SGE activities in FSGF are located in remote areas [10], which makes the residents have a moderate level of knowledge. At the same time, the residents have to depend on the information given by experts and regulators, and a limited number of information sources are also expected to increase trust in scientists and authorities [13].

Table 2 also shows high levels of residents’ perceived benefits and acceptance of SGE in FSGF, with all three indicators above 3 in mean values. This might be attributed to that the local government has acquired a 1% stake in the shale company for enhancing the local infrastructure (e.g., tap water supplies and roads) and community development [13], and a high level of acceptance is also consistent with the previous findings [14,15]. For another endogenous construct, residents’ risk perceptions suggest that their perceptions of environmental risks were higher than that of health risks and basically agreed that the current regulations should be improved to further address SGE’s risks. Overall, our results roughly indicate that residents’ overall benefit perceptions outweighed the risk perceptions, which contributes to the varied findings in comparing the risk and benefit perceptions of SGE [7,18,42]. The reason might be that FSGF has been given particular attention as the nation’s first and largest demonstration area of SGE, which makes the residents have a considerable benefit perception. The effects of pro-shale propaganda, coupled with achievements in local infrastructure construction and economic development, have brought more benefit perceptions for people living in this area.

A confirmatory factor analysis (CFA) has been applied to test the adequacy of each construct’s measurement model [55]. The assessment of each measurement model, following Hair et al. [58] and Kline [57], consists of internal consistency reliabilities (reflected by Cronbach’s α) and convergent validity (reflected by standardized loadings, composite reliabilities, and the average variance extracted). Specifically, as Table 2 presents, Cronbach’s α of all constructs range from 0.793 to 0.883, higher than the recommended level of 0.7 [58]. Of each construct, Table 2 also proves that standardized loadings on all indicators are higher than the recommended level of 0.7, the composite reliability (CR) is higher than the recommended level of 0.7, and the average variance extracted (AVE) is higher than the recommended level of 0.5 [58]. Therefore, results from the CFA indicate that our measurement models have good reliability and validity.

### 4.2. Structural Equation Modeling

The SEM has been used to test the proposed research hypotheses presented in Figure 1. Based on our sample’s covariance matrix, the maximum likelihood estimation of the path coefficients is presented in Table 3 (Table A3 in the Appendix A Materials provides the complete set of results of the SEM).

Before going into detail, we need to assess the goodness-of-fit of the SEM results using various fit statistics. The following fit statistics are obtained from applying the SEM to test our research hypotheses presented in Figure 1. The χ^2^ (*df* = 144) is 378.041, which indicates a significant discrepancy between the sample covariance matrix and the fitted covariance matrix based on the hypothesized relationships. However, Jöreskog and Sörbom [59] argued that the χ^2^ is sensitive to the sample size, and models with a large sample size greater than 400 (which is the case in this study) nearly always reject the hypothesized relationships. Since the χ^2^ test is no longer a reasonable measure of model fitness in our case, other fit indices (χ^2^/df = 2.625, RMSEA = 0.044, GFI = 0.956, AGFI = 0.941, PGFI = 0.724, NFI = 0.950, RFI = 0.941, IFI = 0.968, TLI = 0.962, CFI = 0.968, PNFI = 0.80, and PCFI = 0.815) can prove that our model meets the widely accepted goodness of fit standards [58].

The estimated path coefficients show that perceived risks (*β*_H7_ = −0.205, *p* < 0.001) and perceived benefits (*β*_H8_ = 0.468, *p* < 0.001) have the significant opposite impacts on *Acceptance*. Our model explains 27.4% of the variance of *Acceptance*, and the results confirm that the higher degree of *Acceptance* can be caused by the higher degree of *Benefit* or the lower degree of *Risk*, which is consistent with the theory and other empirical studies [15,40]. Thus, hypotheses H7 and H8 are supported. Moreover, a bootstrapping procedure with 2000 replications is proposed for detecting the influential difference between this pair of paths (with the null hypothesis, *β*_H7_ + *β*_H8_ = 0), and the results show that the 90% bootstrap confidence interval of the influential difference is from 0.151 to 0.383, with *p* = 0.001. Thus, for our surveyed residents, the influence of perceived SGE benefits to local communities is significantly greater than that of perceived SGE risks.

Regarding the mediator construct of *Risk*, our model explains 41.6% of its variance. The estimated path coefficients confirm that *Fairness* (*β*_H1_ = −0.517, *p* < 0.001) and *Trust* (*β*_H5_ = −0.447, *p* < 0.001) are negatively associated with perceived risks, while *Knowledge* (*β*_H3_ = 0.338, *p* < 0.001) is positively associated with perceived risks. Therefore, hypotheses H1, H3, and H5 are supported. The test results from bootstrapping show that *Fairness*, *Trust,* and *Knowledge* have a decreasing sequence of influences on perceived risks. As for another mediator construct of *Benefit*, our model explains 18.4% of its variance. The estimated path coefficients also indicate that the influences of *Fairness* (*β*_H2_ = 0.336, *p* < 0.001) and *Trust* (*β*_H6_ = 0.232, *p* < 0.001) are positive and significant, which supports our hypotheses H2 and H6. The test results from bootstrapping show that *Fairness* has a greater influence on *Benefit* than *Trust*. By contrast, hypothesis H4 is not supported by our data, since there is no significant influence of *Knowledge* on the perceived benefits (*β*_H4_ = 0.012, *p* = 0.660).

The bootstrapping procedure again has been used to estimate each the exogenous construct’s partial indirect effect and total indirect effects on *Acceptance* (see Table A4 in the Appendix A Materials). In-line with the expectations, *Fairness* (*β* = 0.264) and *Trust* (*β* = 0.200) positively and significantly affect *Acceptance*, while *Knowledge* (*β* = −0.064) negatively and significantly affects *Acceptance*. Moreover, Table A4 also shows that *Fairness* is the most important exogenous determinant of *Acceptance*, followed by *Trust* and *Knowledge*.

### 4.3. Discussion

The SEM has been used to examine the influencing factors involved in residents’ acceptance of SGE. Our results indicate that residents who perceive more benefits and fewer risks of SGE will have higher levels of acceptance of local SGE, as hypothesized. The influence of *Benefit* on *Acceptance* is significantly larger than that of *Risk*. This result is similar to the findings from Yu et al.’s [15] survey in another part of Southwest China using a regression model. However, it is inconsistent with the finding of Evensen and Stedman [27], who found that the relationship between risk perception and support is stronger than the relationship between benefit perception and support in the Marcellus Shale region of the US. The discrepancy implies that residents in Southwest China are likely to pay more attention to the benefits brought by SGE facilities to communities. The reason may be attributed to the differences in public concerns between China and the US. Residents in China consider that most of the SGE’s adverse impacts are under control [15] and believe that SGE can deliver multiple benefits to the society and economy [9]. Another possible reason is that SGE in Southwest China are located in less developed areas, and people in these areas tend to care more about the benefits rather than the risks associated with the potentially hazardous facility in their communities [60].

As hypothesized, we also find that all three exogenous constructs have indirect influences on *Acceptance* via the mediators. Overall, *Fairness* has the largest contribution in explaining all three endogenous constructs. Since sustainable and legitimate SGE governance requires acceptance from a variety of stakeholders [32], our findings indicate that residents’ meaningful participation in policy processes is a crucial part of achieving this. Moreover, the influences of *Trust* on all three endogenous constructs have the expected signs. The effect of *Trust* on *Acceptance* substantiates Wallquist et al.’s [51] assertions that trust in the actors related to SGE has an indirect influence on *Acceptance* through risk and benefit perceptions. In other words, residents employ cognitive shortcuts to make conclusions about the acceptance of local SGE and use their trust evaluation in the responsible actors as a heuristic to evaluate the risks and benefits of SGE. Finally, the results also reveal that *Knowledge* is the least important determinant of all three endogenous constructs, and *Knowledge’s* hypothesized relationship with *Benefit* is not significant. A possible explanation for our findings may be caused by the residents’ previous unpleasant experience (e.g., water pollution, earthquake, and air pollution) that could possibly be linked to drilling and fracking activities. For instance, residents who have experienced the SGE’s adverse impacts tend to seek information to support their instincts, which would make these residents more knowledgeable about the possible environmental and health risks posed by SGE. Thus, more knowledge of SGE’s environmental impacts is often less relevant to their perceived benefits while associated with a higher certainty of the risks, eventually leading to lower acceptance.

## 5. Conclusions and Implications

The rapidly expanding SGE in China and its potential impacts on communities call for a better understanding of the residents’ reactions towards local SGE in order to achieve a more supportive and sustainable energy system. A literature-based conceptual model has been suggested for evaluating the effects of residents’ perceived fairness, knowledge, and trust on their acceptance, which, in turn, are fully mediated by their perceived risks and perceived benefits. To examine the causal relationships among these psychological constructs, structural equation modeling has been applied to a cross-sectional dataset consisting of 825 surveyed households in Fuling shale gas field, Southwest China. Following are the main findings, as well as their policy implications and limitations in this study.

This study has some theoretical contributions by providing theoretical insights that help to understand the determinants of residents’ acceptance of local SGE. While the majority of qualitative studies have examined the determinants of the public’s attitudes with the use of general population samples (e.g., [33,34,35]), this study focuses on the decision-making process of people living nearby SGE activities. Moreover, our research framework fits the data well, and results from the SEM largely confirm the hypothesized causal relationships: perceived benefits and perceived risks directly affect residents’ acceptance, while fairness, knowledge, and trust indirectly affect residents’ acceptance through risk and benefit perceptions. Hence, our study enriches the theory and evidence of the energy technology acceptance framework. Further research is also required to include other relevant constructs and establish more detailed relationships among them.

Our findings also show that residents’ perceived benefits have the greatest influence on their acceptance, significantly larger than perceived risks. Hence, the acknowledgment of social and economic benefits to local communities delivered by SGE is likely to increase acceptance significantly. Local authorities and shale companies should prioritize nearby residents with easily perceived benefits, such as by enhancing employability, improving the infrastructure, and sponsoring local events. The influence of risk perception on acceptance, although roughly half of that of benefit perception, should also be taken seriously in policy design, especially considering that proponents’ opinions on energy technology are less stable than those of opponents [61]. Local authorities and shale companies should take necessary measures to reduce residents’ perceived risks posed by SGE, including the introduction of more stringent environmental regulations, the enhancement of local environmental enforcement performances, and the improvement of health risk communications.

Another major insight from our study is that residents’ acceptance is indirectly influenced by their perceived fairness in energy policy decisions, trust in scientists and the central government, and their knowledge of SGE’s environmental impacts. If residents have more fairness and trust perceptions, then fewer risks and more benefits will be perceived, leading to a greater acceptance of SGE. The influences of residents’ perceived fairness and trust highlight the importance of public engagement and actors’ trust to secure the acceptance of SGE. These measures could consist of ensuring the residents would have a legitimate voice to be treated with respect and listened to, taking advantage of the emerging media to send a message about science directly to public audiences, offering programs and incentives for community-based partnerships in SGE (e.g., water quantity and quality monitoring, well site reclamation, and facilities maintenance). Regarding the negative relationship between residents’ knowledge of SGE impacts and their acceptance, specific focus should be placed on improving risk communications with affected communities. These measures could consist of addressing the fears and concerns expressed by the residents, identifying which language and linguistic choices help counteract the residents’ confirmation bias, and eliciting beneficial changes in residents’ knowledge of SGE environmental impacts by helping them understand how science really works.

Overall, our findings underscore the importance of gaining further insight into the psychological processes and mechanisms underpinning residents’ acceptance of local SGE. Despite the findings and implications in this study, it also has several limitations. First, we acknowledge that our conceptual framework may not have considered all the relevant factors to explain residents’ acceptance of local SGE. Research in the factors related to energy technology acceptance indicates that other determinants may also have influences on this concept, such as the influences of affective feeling and values [19,21]. The conceptual framework can also be extended to incorporate the influences of socioeconomic variables to give a more comprehensive explanation [14,15] of residents’ acceptance of local SGE. Second, the items in our questionnaire are measured by residents’ general perceptions, as opposed to measures related to specific SGE activities. Thus, respondents’ self-reported data can contain potential bias, since responses to our general questions could be based on their recollection of specific incidents rather than a thorough consideration of the overall SGE system. Third, our conclusion is based on a site-specific survey of household living in China’s SGE areas. Further research is still needed to explore the conclusion’s validity in different countries (e.g., favor shale countries versus oppose shale countries) and among different groups of respondents (e.g., households versus individuals and local communities versus the general public).

## Figures and Tables

**Figure 1 ijerph-17-06736-f001:**
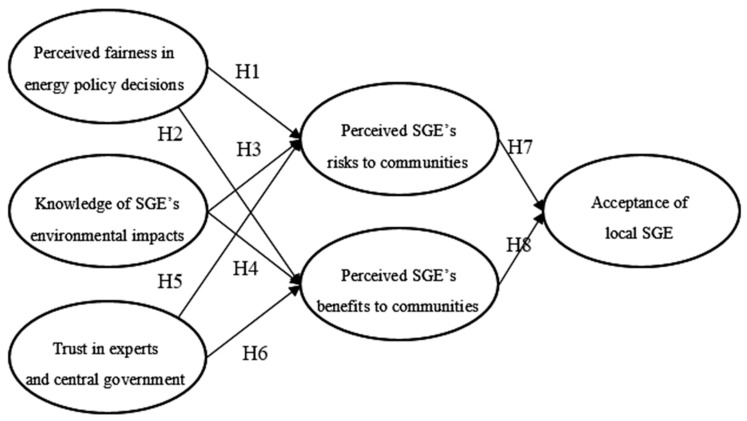
The conceptual process of residents’ acceptance of shale gas exploitation (SGE).

**Figure 2 ijerph-17-06736-f002:**
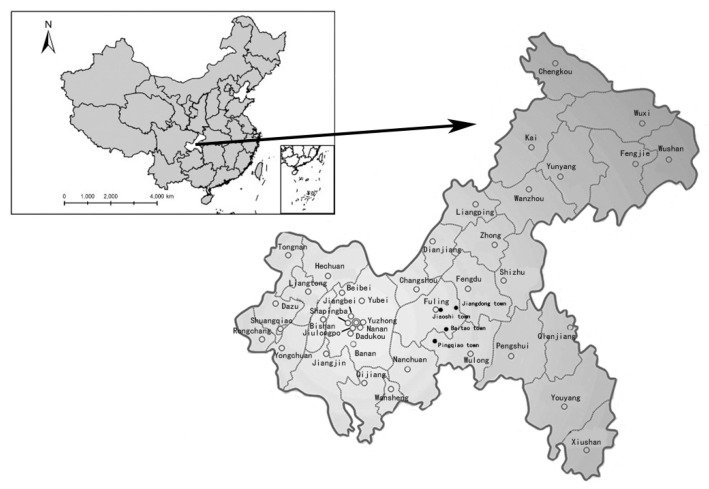
Map of the Chongqing Municipality and studied areas (labeled with •).

**Table 1 ijerph-17-06736-t001:** The measurements of the psychological constructs and socio-demographics. FSGF: Fuling shale gas field.

**Perceived Fairness In Energy Policy Decisions (denoted as *Fairness*, *E1*)**
I think residents who bear the risks of SGE have received proper compensation (*x11*). I feel SGE has a transparent and explicit procedure to mitigate the potential risks (*x12*). I believe residents’ concerns will be taken seriously by decision-makers (*x13*). I know there are effective ways to raise my concerns about SGE (*x14*). Relevant references: King and Murphy [49] and Schlosberg [28].
**Knowledge of SGE’s Environmental Impacts (Denoted as *Knowledge*, *E2*)**
I feel well-informed about the possible environmental impacts of SGE (*x21*). I have got enough information to have an opinion on SGE’s environmental impacts (*x22*). I understand the environmental impacts of SGE in the news (*x23*). Relevant references: Costa-Font and Gil [50] and Costa et al. [35].
**Trust in Scientists and Authorities (Denoted as *Trust*, *E3*)**
I trust the information given by scientists and authorities (*x31*). I have confidence in scientists and authorities that they can adequately assess risk for humans and the environment (*x32*). I believe scientists and authorities would inform the public if a severe danger is posed by SGE (*x33*). Relevant references: Huijts et al. [23] and Wallquist et al. [51].
**Perceived SGE’s Risks to Communities (Denoted as *Risk*, *K1*)**
I am trouble by the environmental impacts, such as water pollution and earthquakes (*y11*). I am worried about the potential hazard to human health (*y12*). I think our current regulations are not sufficient to prevent the risks associated with hydraulic fracturing (*y13*). Relevant references: Brasier et al. [52], Costa et al. [35], and Wallquist et al. [51].
**Perceived SGE’s Benefits to Communities (Denoted as *Benefit*, *K2*)**
I am convinced that SGE provides job and income opportunities for residents (*y21*). I realize that SGE facilitates local infrastructure construction (*y22*). I feel residents’ sense of community pride has been enhanced (*y23*). Relevant references: Brasier et al. [52] and Evensen et al. [53].
**Acceptance of Local SGE (Denoted as *Acceptance*, *K3*)**
I accept SGE in FSGF (*y31*). I agree that SGE activities can continue in the future (*y32*). I support expanding SGE areas in FSGF (*y33*). Relevant references: Evensen et al. [53].
**Socio-Demographics**
Age of the surveyed respondent. Gender of the surveyed respondent. Average personal annual income (with the units of Chinese yuan: less than 3000 = 1, (3000, 6000) = 2, (6000, 9000) = 3, (9000, 12,000) = 4, (12,000, 15,000) = 5, (15,000, 18,000) = 6, and above 18,000 = 7. Education level of the respondent: Primary school or below = 1, junior high school = 2, senior high school = 3, and college degree or above = 4. Household location: first-phase construction area = 1 and second-phase construction area = 0.

**Table 2 ijerph-17-06736-t002:** Confirmatory factor analysis (CFA) of the constructs.

Construct	Indicator	Mean	Std. Dev.	Std. Load.	Assessment
*Fairness* *(E1)*	*x11*	2.297	1.315	0.712	α = 0.839 CR = 0.846 AVE = 0.579
	*x12*	1.750	0.796	0.754	
	*x13*	2.823	1.461	0.798	
	*x14*	3.101	1.477	0.737	
*Knowledge* *(E2)*	*x21*	3.172	1.642	0.844	α = 0.837 CR = 0.840 AVE = 0.637
	*x22*	3.284	1.635	0.84	
	*x23*	2.787	1.618	0.707	
*Trust* *(E3)*	*x31*	3.935	0.919	0.757	α = 0.793 CR = 0.795 AVE = 0.564
	*x32*	3.445	0.981	0.751	
	*x33*	2.427	1.384	0.738	
*Risk* *(K1)*	*y11*	3.314	1.461	0.825	α = 0.855 CR = 0.863 AVE = 0.677
	*y12*	2.964	1.591	0.794	
	*y13*	3.686	1.054	0.769	
*Benefit* *(K2)*	*y21*	3.775	0.802	0.805	α = 0.820 CR = 0.834 AVE = 0.626
	*y22*	4.068	0.999	0.779	
	*y23*	3.076	1.552	0.712	
*Acceptance* *(K3)*	*y31*	4.155	0.853	0.891	α = 0.883 CR = 0.886 AVE = 0.722
	*y32*	3.961	0.920	0.838	
	*y33*	3.778	1.111	0.839	

Notes: Std. Dev., standard deviation; Std. Load., standardized loading; α, Cronbach’s alpha; CR, composite reliability; and AVE, average variance extracted.

**Table 3 ijerph-17-06736-t003:** Results about hypothesis testing in the structural model.

Research Hypothesis	Unstandardized Path Coefficient	Standardized Path Coefficient	*p*-Values	Conclusion
H1: *K1*←*E1*	−0.517	−0.397	0.000	Supported
H2: *K2*←*E1*	0.336	0.347	0.000	Supported
H3: *K1*←*E2*	0.338	0.358	0.000	Supported
H4: *K2*←*E2*	0.012	0.017	0.660	Not Supported
H5: *K1*←*E3*	−0.447	−0.361	0.000	Supported
H6: *K2*←*E3*	0.232	0.251	0.000	Supported
H7: *K3*←*K1*	−0.205	−0.243	0.000	Supported
H8: *K3*←*K2*	0.468	0.413	0.000	Supported

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
