# Peer review of "The Psychological Process of Residents’ Acceptance of Local Shale Gas Exploitation in China"

_ijerph, 2020, doi:10.3390/ijerph17186736_

Round 1
Reviewer 1 Report
The authors did a good job in motivating their study. Specifically, they clearly compare and contrast their study with the findings in the literature. They explain why doing their survey on the opinions of individuals living in the area of the shaded gas exploration. The structural modeling is well executed. The literature is up to date. The writing is smooth.
Author Response
Thank you so much for reading the article carefully and your positive comment on the manuscript.
Reviewer 2 Report
- The author's name is missing from reference 33, please add it. (L 123)
- Please add a picture of the study area so that readers can understand the location of the area.
- In addition to the basic information such as the age and education background of the interviewees in Section 3.3, it is also necessary to emphasize whether the respondents' answers to the questionnaire are affected by subjectivity. For example, some interviewees are familiar with shale gas, while others have no exposure to shale gas, and whether this will affect the questionnaire results.
- The introduction to the structural equation model can be placed separately in section 3 to introduce the specific implementation process of this method in this paper
- In section 4.1, the means of fairness, benefit and Acceptance have been explained, but there are no explanation about the means of Knowledge, Trust and Risk, please explain.
- How was the data obtained in section 4.2? eg, The χ2 353 (df = 144) is 378.041, It seems that these data are not the ones in Table 3. Please explain. (L352-360)
- The discussion section should be added to carry out the discussion according to the experimental results. The discussion should focus on issues related to the results, highlight the innovation and importance of the research, and compare the results with relevant research results.
Author Response
Thank you so much for reading the article carefully and for giving us detailed comments on it. We have attended each comment and edited the article accordingly. The point-by-point responses to your comments are as follows.
Comment 1: The author's name is missing from reference 33, please add it. (L 123).
Response: We agree and the revised manuscript has been improved accordingly.
Comment 2: Please add a picture of the study area so that readers can understand the location of the area.
Response: We agree and a map of the study area has been added to the revised manuscript in Section 3.1.
Comment 3: In addition to the basic information such as the age and education background of the interviewees in Section 3.3, it is also necessary to emphasize whether the respondents' answers to the questionnaire are affected by subjectivity. For example, some interviewees are familiar with shale gas, while others have no exposure to shale gas, and whether this will affect the questionnaire results.
Response: We thank the reviewer for this important comment.
The psychological process of acceptance is reflected by several literature-based subjective questions in this study. We believe that some exogenous variables, e.g. familiarity with shale gas, may have influences on the questionnaire results. Nevertheless, our research purpose is to examine the psychological determinants of residents’ acceptance (as indicated by the title of this study). The influences of socio-economic variables will be left for further studies. To address the reviewer’s concern, the revised paper now provides the following discussion in the limitation section (Section 5).
“First, we acknowledge that our conceptual framework may not have considered all the relevant factors to explain residents’ acceptance of local SGE. Research in the factors related to energy technology acceptance indicates that other determinants may also have influences on this concept, such as the influences of affective feeling and values (c.f. [19, 21]). The conceptual framework can also be extended to incorporate the influences of socio-economic variables to give a more comprehensive explanation (c.f. [14-15]) of residents’ acceptance of local SGE.”
Comment 4: The introduction to the structural equation model can be placed separately in section 3 to introduce the specific implementation process of this method in this paper.
Response: We agree with the reviewer that the readers may be unfamiliar with the structural equation model. Nevertheless, the method (structural equation model) used in this paper is exactly the same with that of other studies using structural equation model.
Since this study doesn’t aim at methodology innovation, we do not include this part in our manuscript. As the other SEM studies (e.g. Huijts and Molin (2014)), we have provided an illustration of the model specification by Figure 1 in Section 2. At the same time, if the readers are interested in the specific form of the model, we have added a reference to the introduction of the structural equation model within the revised manuscript.
“Structural equation modeling (SEM) (see Anderson & Gerbing [50] for a review) has been used to test the research hypotheses, as well as the reliability and validity of the measurement in this study.”
Comment 5: In section 4.1, the means of fairness, benefit and Acceptance have been explained, but there are no explanation about the means of Knowledge, Trust and Risk, please explain.
Response: We agree. Firstly, we have provided the explanation about the means of Knowledge and Trust at the end of second paragraph in Section 4.1:
“This might because that most of the SGE activities in FSGF is located in remote areas [10], which makes the residents have a moderate level of knowledge. At the same time, the residents have to depend on the information given by experts and regulators, and a limited number of information sources are also expected to increase trust in scientists and authorities [13].”
Secondly, the explanation about the means of Risk in the third paragraph in Section 4.1:
“The reason might be that FSGF has been given particular attention as the nation’s first and largest demonstration area of SGE, which makes the residents have a considerable benefit perception. The effects of pro-shale propaganda, coupled with achievements in local infrastructure construction and economic development, have brought more benefit perception for people living in this area.”
Comment 6: How was the data obtained in section 4.2? eg, The χ2 353 (df = 144) is 378.041, It seems that these data are not the ones in Table 3. Please explain. (L352-360).
Response: Yes, these data were not presented in Table 3. These data were obtained from using the SEM to test our research hypotheses presented in Figure 1. For the sake of brevity, typical SEM studies usually do not include the fit statistics in the table, e.g. Costa-Font and Gil (2009), Huijts and Molin (2014), and Tang et al. (2013). Our results show typical patterns associated with the SEM literature, in which the goodness-of-fit statistics are presented only within the text. As noted in the revised paper, we have made this more clear in Section 4.2:
“Before going into detail, we need to assess the goodness-of-fit of the SEM results using various fit statistics. The following fit statistics are obtained from applying SEM to test our research hypotheses presented in Figure 1.”
Comment 7: The discussion section should be added to carry out the discussion according to the experimental results. The discussion should focus on issues related to the results, highlight the innovation and importance of the research, and compare the results with relevant research results.
Response: We agree and the following discussion part has been added to the revised manuscript in Section .
“The SEM has been used to examine the influencing factors involved in residents’ acceptance of SGE. Our results indicate that residents who perceive more benefits and fewer risks of SGE will have higher levels of acceptance of local SGE, as hypothesized. The influence of Benefit on Acceptance is significantly larger than that of Risk. This result is similar to the findings from Yu et al.’s [15] survey in another part of southwest China using a regression model. However, it is inconsistent with the finding of Evensen & Stedman [27], who found that the relationship between risk perception and support is stronger than the relationship between benefit perception and support in the Marcellus Shale region of the US. The discrepancy implies that residents in southwest China are likely to pay more attention to the benefits brought by SGE’s facilities to communities. The reason may be attributed to the differences in public concerns between China and the US. Residents in China consider that most of the SGE’s adverse impacts are under control [15], and believe that SGE can deliver multiple benefits to the society and economy [9]. Another possible reason is that, SGE in southwest China is located in less development areas, and people in these areas tend to care more about the benefits rather than the risks associated with the potentially hazardous facility in their communities [57].
As hypothesized, we also find that all three exogenous constructs have indirect influences on Acceptance via the mediators. Overall, Fairness has the largest contribution in explaining all three endogenous constructs. Since sustainable and legitimate SGE governance requires acceptance from a variety of stakeholders [32], our finding indicates that residents’ meaningful participation in policy processes is a crucial part of achieving this. Moreover, the influences of Trust on all three endogenous constructs have the expected signs. The effect of Trust on acceptance substantiates Wallquist et al. ’s [53] assertion that trust in the actors related to SGE has an indirect influence on acceptance through risk and benefit perceptions. In other words, residents employ cognitive shortcuts to make conclusions about the acceptance of local SGE, and use their trust evaluation in the responsible actors as a heuristic to evaluate the risks and benefits of SGE. Finally, the results also reveal that Knowledge is the least important determinant of all three endogenous constructs, and Knowledge’s hypothesized relationship with Benefit is not significant. A possible explanation for our findings may be caused by the residents’ previous unpleasant experience (e.g., water pollution, earthquake, air pollution) that could possibly be linked to the drilling and fracking activities. For instance, residents who have experienced the SGE’s adverse impacts tend to seek information to support their instincts, which would make these residents more knowledgeable about the possible environmental and health risks posed by SGE. Thus, more knowledge of SGE’s environmental impacts is often less relevant to their perceived benefits while associated with a higher certainty of the risks, eventually leading to lower acceptance.”
We again thank the reviewer for the time spent reading our manuscript and the extremely helpful comments. These have been invaluable in helping to improve the manuscript.
References:
Huijts, N. M., Molin, E. J., & van Wee, B. (2014). Hydrogen fuel station acceptance: A structural equation model based on the technology acceptance framework. Journal of Environmental Psychology, 38, 153-166.
Costa-Font, M., & Gil, J. M. (2009). Structural equation modelling of consumer acceptance of genetically modified (GM) food in the Mediterranean Europe: A cross country study. Food Quality and Preference, 20(6), 399-409.
Tang, J., Folmer, H., & Xue, J. (2013). Estimation of awareness and perception of water scarcity among farmers in the Guanzhong Plain, China, by means of a structural equation model. Journal of environmental management, 126, 55-62.
Reviewer 3 Report
The authors present the results of a survey on Shale Gas Exploitation in China. It is overall well-designed and written. There are some specific grammatical and mechanistic errors, but conceptually there are no significant concerns. Specific corrections are listed below:
Lines 46, 51: The authors use "Western" and "western". It should be standardized to "Western".
Line 79: The authors should list the references sequentially: [7,18]
Line 163: "irrelevant with" should be changed to "unrelated to" or a similar phrase.
Line 169: In this phrase: "levels between the central and" remove "the".
Lines 183-184: Rewrite to include "gas" in "greenhouse gas emissions".
Line 186: Check the numbering of all your references. Here Sovacool is listed as [45] but it is actually [44] in the references section.
Line 193: The references are in the wrong order and should read [14, 42]
Line 206: "filed" should be "field"
Line 211 (and elsewhere): Superscript "2" in "km2"
Line 213: "planned to end" does not make sense. The authors need to include "is" in front of "planned"
Table 1: What are the personal income units? The authors should include the equivalent in USD or EUR
Line 260: "representative" should be "respresentativeness"
Lines 266-267: "simple random sample...was randomly selected" is too repetitive
Line 436: "process" and "mechanism" should be plural
Author Response
Thank you so much for reading the article carefully and for giving us detailed comments on it. We have attended each comment and edited the article accordingly. The point-by-point responses to your comments are as follows.
Comment 1: Lines 46, 51: The authors use "Western" and "western". It should be standardized to "Western".
Response: We agree and the revised manuscript has been improved accordingly.
Comment 2: Line 79: The authors should list the references sequentially: [7,18].
Response: We agree and have made the correction of the citation.
Comment 3: Line 163: "irrelevant with" should be changed to "unrelated to" or a similar phrase.
Response: We agree and the revised manuscript has been improved accordingly.
Comment 4: Line 169: In this phrase: "levels between the central and" remove "the".
Response: We agree and “the” has been removed in the revised manuscript.
Comment 5: Lines 183-184: Rewrite to include "gas" in "greenhouse gas emissions".
Response: We agree and “greenhouse emissions” has been revised to “greenhouse gas emissions” in the revised manuscript.
Comment 6: Line 186: Check the numbering of all your references. Here Sovacool is listed as [45] but it is actually [44] in the references section.
Response: We agree. This mistake has been caused by the missing of two references in the references section. We have revised manuscript accordingly to make sure that all cited references have been mentioned in the text and vice versa.
Comment 7: Line 193: The references are in the wrong order and should read [14, 42].
Response: We agree and the revised manuscript has been improved accordingly.
Comment 8: Line 206: "filed" should be "field".
Response: We agree and the revised manuscript has been improved accordingly.
Comment 9: Line 211 (and elsewhere): Superscript "2" in "km2".
Response: We agree and the revised manuscript has been improved accordingly.
Comment 10: Line 213: "planned to end" does not make sense. The authors need to include "is" in front of "planned".
Response: We agree and the revised manuscript has been improved accordingly.
Comment 11: Table 1: What are the personal income units? The authors should include the equivalent in USD or EUR.
Response: We agree and the unit of annual income (RMB) has been added.
Comment 12: Line 260: "representative" should be "representativeness". respresentativeness
Response: We agree and the revised manuscript has been improved accordingly.
Comment 13: Lines 266-267: "simple random sample...was randomly selected" is too repetitive.
Response: We agree and this sentence has been revised to “simple random sample...was selected.”
Comment 14: Line 436: "process" and "mechanism" should be plural.
Response: We agree and the revised manuscript has been improved accordingly.
We again thank the reviewer for the time spent reading our manuscript and the extremely helpful comments. These have been invaluable in helping to improve the manuscript.